# Distribution Free Prediction Sets for Node Classification

**Jase Clarkson**
Department of Statistics, University of Oxford
`jason.clarkson@stats.ox.ac.uk`

## Abstract

Graph Neural Networks (GNNs) are able to achieve high classification accuracy on many large real world datasets, but provide no rigorous notion of predictive uncertainty. We leverage recent advances in conformal prediction to construct prediction sets for node classification in inductive learning scenarios, and verify the efficacy of our approach across standard benchmark datasets using popular GNN models. The code is available at this link.

## 1 Introduction

Machine learning on graph structured data has seen a boom of popularity in recent years, with applications ranging from recommendation systems to biology and physics. Graph neural networks are quickly maturing as a technology; many state of the art models are commoditised in frameworks such as Pytorch Geometric [1] and DGL [2]. Despite their overwhelming popularity and success, very little progress has been made towards quantifying the uncertainty of the predictions made by these models, a vital step towards robust real world deployments.

In related areas of machine learning such as computer vision, conformal prediction [3] has emerged as a promising candidate for uncertainty quantification [4]. Conformal prediction is a very appealing approach as it is compatible with any black box machine learning algorithm and dataset as long as the data is statistically exchangeable. The most wide-spread method, so called *split-conformal*, also requires trivial computational overhead when compared to model fitting.

Graph structured data is in general not exchangeable and so the guarantees provided by conformal prediction in its naive form do not hold. Recent work by Barber et al. [5] extends conformal prediction to the non-exchangeable setting and provides theoretical guarantees on the performance of conformal prediction in this setting. We leverage insights from [5] to apply conformal prediction in the node classification setting. The key insight is that for a homophilous graph, the model calibration should be similar in a neighbourhood around any given node. We leverage this insight to localise the calibration of conformal prediction. We show that our method improves calibration of predictive uncertainty and provides tighter prediction sets when compared with a naive application of conformal prediction across several state of the art models applied to popular node classification datasets.

## 2 Conformal Prediction

Conformal prediction is a family of algorithms that generate finite sample valid prediction intervals or sets from an arbitrary black box machine learning model. Conformal prediction may be thought of as a "wrapper" around a fitted model that uses a set of exchangeable held out data to calibrate prediction sets. Amazingly, the predictive model does not even need be well specified for these guarantees to hold (although the prediction intervals or sets may not be useful in this case). In the exposition below we will focus on conformal classification as that is the object of study in this work, but note that conformal prediction can be used for regression and other risk control procedures. We recommend consulting the excellent tutorial by Angelopoulos and Bates [6] for an introduction.

### 2.1 The Exchangeable Case

Suppose we are working on a $K-$class classification problem and we have a fitted model $\hat{f} : \mathcal{X} \to [0, 1]^K$ that outputs the probability of each class. Given an exchangeable set of held-out calibration

J. Clarkson, Distribution Free Prediction Sets for Node Classification (Extended Abstract). Presented at the First Learning on Graphs Conference (LoG 2022), Virtual Event, December 9–12, 2022.

datapoints $(X_1, Y_1), \ldots, (X_n, Y_n)$ (held out meaning they were not used to fit the model) and a new evaluation point $(X_{n+1}, Y_{n+1})$, conformal prediction constructs a *prediction set* $\mathcal{T}(X_{n+1})$ that satisfies

$$1 - \alpha \leq \mathbb{P}\left(Y_{n+1} \in \mathcal{T}(X_{n+1})\right) \leq 1 - \alpha + \frac{1}{n+1} \tag{1}$$

for a user specified error rate $\alpha \in [0, 1]$. Conformal prediction relies on a *score function* $S : \mathcal{X} \times \mathcal{Y} \to \mathbb{R}$, which is a measure of the calibration of the prediction at a given datapoint. Given a score function $S$, the procedure for constructing a prediction set is very simple; for each datapoint $(X_i, Y_i)$ in the calibration set, compute the score $s_i = S(X_i, Y_i)$. Define $1 - \hat{\alpha}$ to be the $\lceil (n + 1)(1 - \alpha) \rceil / n$ empirical quantile of the scores $s_1, \ldots, s_n$, and finally create the prediction set $\mathcal{T}(X_{n+1}) = \{y : S(X_{n+1}, y) \leq 1 - \hat{\alpha}\}$.

A popular conformal prediction procedure for classification problems is known as Adaptive Prediction Sets (APS, [7]). To motivate the APS score function, suppose we have access to an *oracle* classifier $\pi$ that exactly matches the true conditional distribution (i.e. $\pi(x) = \mathbb{P}(Y_{n+1}|X_{n+1} = x)$). Then to construct a $1 - \alpha$ prediction set from the oracle, we simply sort the probabilities into descending order, and add labels to the set until the cumulative probability exceeds $1 - \alpha$ (with appropriate tie breaking to ensure exact coverage).

Let $\{\pi_{(1)}, \ldots, \pi_{(K)}\}$ be the order statistics of the conditional probabilities $\pi(x)$ so that $\pi_{(1)} \geq \pi_{(2)} \geq \cdots \geq \pi_{(K)}$. Prediction sets can be constructed from the oracle as

$$\left\{\pi_{(1)}, \ldots, \pi_{(k)}\right\}, \text{where } k = \inf\left\{k' : \sum_{j=1}^{k'} \pi_{(j)} \geq 1 - \alpha\right\}.$$

In practice the probabilities given by a fitted classifier $\hat{f}(x)$ will usually not be exactly equal to $\mathbb{P}(Y_{n+1}|X_{n+1} = x)$. APS instead measures the deviation from the oracle procedure required to achieve the desired level of coverage on the calibration data; the conformal score is defined as

$$S(x, y) = \sum_{j=1}^{k} \hat{f}(x)_{(j)}, \text{ where } y = k. \tag{2}$$

To give a concrete example, suppose we want to construct prediction sets that contain the true label $90\%$ of the time (so $\alpha = 0.1$). It could be the case that, on our held out data, if we simply add up the ordered softmax outputs until their cumulative sum exceeds $0.9$ we actually get $85\%$ coverage, due to the model being miss-calibrated. Using APS we might calculate that if we construct prediction sets using $1 - \hat{\alpha} = 0.94$ we get $90\%$ coverage, and by exchangeability this translates to $90\%$ coverage on any new test point. We would therefore use the level $\hat{\alpha} = 0.06$ to construct our new prediction sets. Note to get exact coverage ties need to be broken randomly when including the final label in the set, see Appendix A.1 for details.

## 2.2 Beyond Exchangeability

Conformal prediction in the form presented above relies on the assumption that the data points $Z_i = (X_i, Y_i)$ are exchangeable. The exchangeable form of conformal prediction provides no guarantee if these assumptions are violated, however *non-exchangeable conformal prediction* was introduced in the pioneering work of Barber et al. [5].

Formally, the non-exchangeable conformal prediction procedure assumes a choice of deterministic fixed weights $w_1, \ldots, w_n \in [0, 1]$ (normalized as detailed in [5]). As before, one computes the scores $s_1, \ldots, s_n$ but now defines the prediction set in terms of the *weighted quantiles* of the score distribution

$$\widehat{C}_n(X_{n+1}) = \left\{y \in \mathcal{Y} : S(X_{n+1}, y) \leq Q_{1-\alpha}\left(\sum_{i=1}^{n} w_i \cdot \delta_{s_i} + w_{n+1} \cdot \delta_{+\infty}\right)\right\} \tag{3}$$

where $Q_\tau(\cdot)$ denotes the $\tau$-quantile of a distribution and $\delta_x$ denotes a point mass at $x$. Non-exchangeable conformal prediction also comes with performance guarantees; the authors define the *coverage gap*

$$\text{Coverage gap } = (1 - \alpha) - \mathbb{P}\left\{Y_{n+1} \in \widehat{C}_n(X_{n+1})\right\} \tag{4}$$

as the loss of coverage when compared to the exchangeable setting, and show that this can be bounded as follows: let $Z = ((X_1, Y_1), \dots, (X_{n+1}, Y_{n+1}))$ be the full dataset and define $Z^i$ as the same dataset after swapping the test point and the $i^{th}$ training point

$$Z^i = ((X_1, Y_1), \dots, (X_{i-1}, Y_{i-1}), (X_{n+1}, Y_{n+1}), (X_{i+1}, Y_{i+1}), \dots, (X_n, Y_n), (X_i, Y_i)).$$

Then the coverage gap in Equation (4) can be bounded as (Theorem 2a, Barber et al. [5]):

$$\text{Coverage gap} \leq \frac{\sum_{i=1}^n w_i \cdot \mathrm{d}_{TV}\left(Z, Z^i\right)}{1 + \sum_{i=1}^n w_i} \tag{5}$$

where $\mathrm{d}_{TV}$ is the total variation distance. To make this bound small one would like to place a large weight $w_i$ on datapoints that are drawn from a similar distribution to the test point $(X_{n+1}, Y_{n+1})$.

## 3 Conformal Prediction for Node Classification

Consider now the node classification setting: we are given a graph $G = (V, E)$, and for each node $i \in V$ we are given a node feature vector $X_i \in \mathbb{R}^F$ and a label $Y_i \in \mathcal{Y}$. A standard pipeline for node classification usually consists of a GNN model that produces a node embedding $h_i \in \mathbb{R}^H$ followed by a classifier $f : \mathbb{R}^H \to \mathcal{Y}$. Here the data points $Z_i = (X_i, Y_i)$ are certainly not assumed to be exchangeable; the underlying principle of GNN models is that the adjacency matrix of $G$ provides information about the dependency between datapoints (and hence neighbourhood information of $G$ is aggregated and used for prediction). Barber et al. [5] show in particular that non-exchangeable data can be navigated when the fitted model is a symmetric function of the test data. Our method is based on the observation that using only training data to fit the model trivially satisfies this assumption. In particular, this excludes the transductive setting.

We combine non-exchangeable conformal prediction with the information given by the adjacency matrix into an algorithm for constructing prediction sets for node classification, which we call Neighbourhood Adaptive Prediction Sets (NAPS). We set the weights in Equation (3) to $w_i = 1$ if $i \in \mathcal{N}_{n+1}^k$, where $\mathcal{N}_{n+1}^k$ is the $k$-hop neighbourhood of node $n + 1$. We then apply non-exchangeable conformal prediction with the APS scoring function in Equation (2). The coverage gap of NAPS is bounded as

$$\text{Coverage gap} \leq \frac{\sum_{i \in \mathcal{N}_{n+1}^k} \mathrm{d}_{TV}\left(Z, Z^i\right)}{1 + |\mathcal{N}_{n+1}^k|} \tag{6}$$

by simple substitution into Equation (5). This bound will be small if the $k$-hop neighbours of node $n + 1$ are distributed similarly, which is otherwise known as *homophily* [8]. Homophily is a key principle of many real world networks, where linked nodes often belong to the same class and have similar features, and is in crucial for good performance in many popular GNN architectures (although recent work has considered the heterophilic case, see [9], which we will discuss in the future work section). This is also related to network *homogeneity*, where nodes in a neighbourhood play similar roles in the network and are considered interchangeable on average.

Note that NAPS is not applicable in the transductive setting, as the fitted model $\hat{f}$ would depend on the node features in the test set, hence the conformal scores would no longer be exchangeable. It is however applicable in inductive settings where either the test set consists of multiple new graphs, or new nodes are added to an existing network. The neighbourhood depth parameter $k$ introduces a tradeoff; expanding the neighbourhood increases the sample size for calibration, but introduces nodes that may be progressively less exchangeable with the test node. In the form presented here we recommend only applying NAPS to large homophilous networks with dense 1 or 2 hop neighbourhoods, but we will discuss extensions in future work.

## 4 Experiments

We now perform experiments with popular real world datasets and models to evaluate the performance of our procedure. Our experiments follow the following format: we split each graph into training, validation and test nodes (where the validation and test nodes are not available during model fitting i.e. an inductive node split). The training and validation sets are used for model fitting, and the test set is used to evaluate the conformal prediction procedure by constructing prediction sets and evaluating the empirical coverage.

## 4.1 Evaluating Conformal Prediction

The observed coverage in a single application of conformal prediction is a *random* quantity, where the randomness comes from the choice of which data points are used for calibration as well as the finite sample size of the calibration set (corresponding to the upper bound in Equation (1)). It is therefore important to pick a large enough number of calibration points, and also repeat the experiment many times with different calibration/evaluation splits. For simplicity we follow the guidelines given in Angelopoulos and Bates [6], which suggest using at least 1000 validation points, and we repeat each experiment 100 times with a different calibration/evaluation split; with this setup by the law of large numbers the probability of observing significant deviations from the true coverage is extremely low, and therefore we can evaluate the performance of our method with high confidence.

Conformal prediction in the exchangeable setting is usually deployed by splitting the data into a calibration set and an evaluation set. The calibration points are used to estimate the quantile of the score distribution, which is used to construct prediction sets for each evaluation point. In our setting, this corresponds to selecting the calibration and evaluation nodes randomly, which ignores the graph structure. The goal of our experimental setup is to study the improvement in the performance of conformal prediction when the graph structure is taken into account.

## 4.2 Experimental Setup

In each experiment, we sample a batch of evaluation nodes and construct a $1 - \alpha$ probability prediction set for each evaluation node using NAPS as described in Section 3, as well as using a naive application of APS calibrated among all the nodes not in the evaluation set. We then report the empirical coverage, average prediction set size and average size of the prediction set given that the set contains the true label across all nodes. For each experiment we sample 1000 nodes randomly from the nodes in the test set, and we perform 100 repetitions of the experiment (see Appendix 4.1 for a justification of this approach). We only apply our method to large connected components from the test set following the discussion in Section 3 (see Appendix A.2 for details on the datasets and the test set construction procedure). We apply our method to three popular node classification datasets, namely Reddit2 and Flickr introduced in [10] and Amazon Computers introduced in [11]. We apply two variants of two popular GNN models, namely GraphSAGE [12] with the mean and max aggregators, and the ShaDow [13] subgraph sampling scheme with GraphSAGE and GCN [14] layers. The results are displayed in Tables 1, 2 and 3 respectively. We see across all models on all three datasets, NAPS produces well calibrated, tight prediction sets, while the naive application of APS tends to overcover and produces wider prediction sets.

**Table 1:** The test accuracy, empirical coverage, average prediction set size and average prediction set size conditional on coverage for all models considered on the Reddit2 dataset with $\alpha = 0.1$. Bold indicates the best performing method.

| Model | Accuracy | Coverage | | Size | | Size \| Coverage | |
|---|---|---|---|---|---|---|---|
| | Top-1 | APS | NAPS | APS | NAPS | APS | NAPS |
| GraphSAGE-Mean | 0.914 | 0.928 | 0.897 | 2.23 | **1.77** | 2.37 | **1.93** |
| GraphSAGE-Max | 0.771 | 0.918 | 0.904 | 3.97 | **3.41** | 4.08 | **3.53** |
| ShaDow-SAGE | 0.844 | 0.930 | 0.902 | 2.15 | **1.72** | 2.21 | **1.78** |
| ShaDow-GCN | 0.827 | 0.931 | 0.902 | 2.18 | **1.73** | 2.22 | **1.81** |

**Table 2:** The test accuracy, empirical coverage, average prediction set size and average prediction set size conditional on coverage for all models considered on the Flickr dataset with $\alpha = 0.1$.

| Model | Accuracy | Coverage | | Size | | Size \| Coverage | |
|---|---|---|---|---|---|---|---|
| | Top-1 | APS | NAPS | APS | NAPS | APS | NAPS |
| GraphSAGE-Mean | 0.503 | 0.912 | 0.904 | 4.22 | **3.82** | 4.26 | **3.87** |
| GraphSAGE-Max | 0.501 | 0.907 | 0.902 | 4.26 | **4.03** | 4.28 | **4.08** |
| ShaDow-SAGE | 0.500 | 0.910 | 0.904 | 4.24 | **4.02** | 4.25 | **4.09** |
| ShaDow-GCN | 0.496 | 0.913 | 0.905 | 4.25 | **4.05** | 4.26 | **4.01** |

**Table 3:** The test accuracy, empirical coverage, average prediction set size and average prediction set size conditional on coverage for all models considered on the Amazon Computers dataset with $\alpha = 0.1$.

| Model | Accuracy | Coverage | | Size | | Size \| Coverage | |
|---|---|---|---|---|---|---|---|
| | Top-1 | APS | NAPS | APS | NAPS | APS | NAPS |
| GraphSAGE-Mean | 0.854 | 0.905 | 0.902 | 1.50 | **1.44** | 1.57 | **1.50** |
| GraphSAGE-Max | 0.765 | 0.902 | 0.902 | 2.15 | **1.99** | 2.17 | **2.04** |
| ShaDow-SAGE | 0.815 | 0.912 | 0.904 | 1.77 | **1.65** | 1.81 | **1.72** |
| ShaDow-GCN | 0.822 | 0.911 | 0.904 | 1.75 | **1.62** | 1.83 | **1.74** |

## 5  Conclusion and Future Work

In this work we have introduced NAPS, an approach for constructing prediction sets on graph structured data. Our method comes with theoretical guarantees on the coverage and we have shown that our approach produces high quality prediction sets when using popular GNN models on standard node classification datasets. Several natural extensions to NAPS will follow in future work; here we applied equal weights to the scores at each neighbourhood depth, but for a homophilous network one could place more weight on shallower neighbours relative to deeper neighbours. We also only used a fixed neighbourhood size of $k = 2$ (which was chosen without any tuning). Intuitively one would like to select $k$ such that the calibration nodes are "local" within the network while providing a large enough sample size to accurately estimate the quantile $1 - \hat{\alpha}$. It would be useful to study the interplay between the optimal choice of $k$ and the diameter of the network. Our method could also be extended to heterophilic networks; in a heterophilic network nodes tend to be connected to dis-similar nodes. One could therefore calibrate among alternating neighbourhoods $\bigcup_{j=1}^{k} \mathcal{N}_{n+1}^{2j} \backslash \mathcal{N}_{n+1}^{2j-1}$.

NAPS may produce wide prediction sets when deployed on low density networks as the sample size for conformal calibration will be small, see Equation (1). An approach for conformal prediction in hierachical models was introduced in Dunn et al. [15], where the quantiles are calibrated in different groups before being pooled. An approach similar to this could be applied for nodes in small connected components, where calibration on similar neighbourhoods or components could be pooled to provide a better estimate of the conformal quantile.

## Acknowledgements

The author would like to thank Gesine Reinert and Stefanos Bennett for their feedback on earlier versions of this work. The author is supported by the EPSRC CDT in Modern Statistics and Statistical Machine Learning (EP/S023151/1) and the Alan Turing Institute (EP/N510129/1).

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

# A   Appendix

## A.1   Adaptive Prediction Sets

Here we give a formal description of the APS [7] procedure for completeness. Denote the oracle classifier $\pi_y(x) = \mathbb{P}[Y = y \mid X = x]$ for all $y \in \mathcal{Y}$, and again let $\pi_{(1)}(x) \geq \pi_{(2)}(x) \geq \ldots \geq \pi_{(K)}(x)$ be the order statistics of this classifier. For any $\tau \in [0, 1]$ define the generalised conditional quantile as

$$L(x; \pi, \tau) = \min \left\{ k \in \{1, \ldots, K\} : \pi_{(1)}(x) + \pi_{(2)}(x) + \ldots + \pi_{(k)}(x) \geq \tau \right\}. \tag{7}$$

One can now define the set valued function

$$\mathcal{S}(x, u; \pi, \tau) = \begin{cases} \text{Labels of the } L(x; \pi, \tau) - 1 \text{ largest } \pi_y(x), & \text{if } u \leq V(x; \pi, \tau), \\ \text{Labels of the } L(x; \pi, \tau) \text{ largest } \pi_y(x), & \text{otherwise} \end{cases} \tag{8}$$

where

$$V(x; \pi, \tau) = \frac{1}{\pi_{(L(x;\pi,\tau))}(x)} \left[ \sum_{c=1}^{L(x;\pi,\tau)} \pi_{(c)}(x) - \tau \right]. \tag{9}$$

The oracle prediction set may then be defined as

$$C_\alpha^{\text{oracle}}(x) = \mathcal{S}(x, U; \pi, 1 - \alpha)$$

where $U \sim \text{Uniform}(0,1)$ is independent of everything else. The above is saying one should break ties proportional to the gap between the cumulative sum of the ordered probabilities until the true label is included and the desired level $\tau$.

## A.2 Dataset Details

For the experiments above we used the Flickr and Reddit2 datasets from [10], and the Amazon Computers dataset introduced in [11].

The Flickr dataset is constructed using images uploaded to the Flickr site, where the node features consist of the meta-data for each image and the label is the image tag. The Reddit2 dataset is constructed from posts on the social media site Reddit, with posts representing nodes. The node features are bag-of-word vectors from the post, and the label is the community (or sub-reddit) that the post belongs to. The Amazon Computers dataset consists of segments of an Amazon co-purchase graph, where nodes represent goods and links are added between nodes if they are frequently bought together.

Our train/validation/test splits for Flickr and Reddit2 were done using the splits given in the original papers (which are conveniently implemented in Pytorch Geometric [1]). For Amazon Computers we constructed our own split, using 752 nodes for training, 1000 for validation and the remaining 12000 for testing. As mentioned in the main text we tested our graph only on large connected components, which we chose as nodes with at least 50 2-hop neighbours in Flickr and Amazon Computers, and nodes with at least 1000 2-hop neighbours in Reddit2. We call this set of nodes $\mathcal{N}^{cal}$, and report the sizes of these sets as well as some summary statistics about each dataset in Table 4.

NAPS relies on node homophily to minimize the coverage gap bound in Equation (5). Here we verify here that each of these networks is homophilous. We measure this via the node homophily ratio defined in [16] as

$$H = \frac{1}{|\mathcal{V}|} \sum_{v \in \mathcal{V}} \frac{|\{(w, v) : w \in \mathcal{N}(v) \wedge y_v = y_w\}|}{|\mathcal{N}(v)|}.$$

We define a homophilous network as one that has node homophily ratio larger than expected under a random assignment of labels. For a network with $K$ classes, assume each node is assigned class $k$ independently with probability $p_k$. Then for any $(v, w) \in \mathcal{V}$, we have

$$\mathbb{P}(y_v = y_w) = \sum_{k=1}^{K} p_k^2.$$

It follows that the expected homophily ratio under random class assignment is

$$\mathbb{E}[H] = \frac{1}{|\mathcal{V}|} |\mathcal{V}| \sum_{k=1}^{K} p_k^2 = \sum_{k=1}^{K} p_k^2.$$

We report both the observed homophily ratio $\hat{H}$ computed over the induced subgraph of the test nodes for each network as well as the expected node homophily under a random assignment of the labels $H_{rand}$, using the relative node label frequencies as the probabilities $p_k$. We see that Reddit2 and Amazon Computers are strongly homophilous, while Flickr is relatively weakly so.

**Table 4:** Statistics for the Flickr, Reddit2 and Amazon Computers datasets.

| Dataset | Nodes | Edges | # Feat | # Classes | # Test Nodes | $\left|\mathcal{N}^{cal}\right|$ | $\hat{H}$ | $H_{rand}$ |
|---|---|---|---|---|---|---|---|---|
| Flickr | 89,250 | 899,756 | 500 | 7 | 22313 | 5161 | 0.319 | 0.266 |
| Reddit2 | 232,965 | 23,213,838 | 602 | 41 | 55334 | 22160 | 0.812 | 0.051 |
| Amazon Comp. | 13752 | 491,722 | 767 | 10 | 12000 | 11033 | 0.785 | 0.208 |

## A.3 Model Training Details

We used the implementations of GraphSAGE and ShaDow provided by Pytorch Geometric [1]. All models on all datasets used the same hyper-parameters. Each GNN used 2 layers with hidden dimension $H = 64$. We used the Adam optimiser [17] with default hyper-parameters, learning rate $\eta = 0.1$, and used dropout probability $\delta = 0.5$. For the GraphSAGE neighbour sampling training we used 25 1-hop neighbours and 10 2-hop neighbours. We used early stopping based on the accuracy on the validation set. We made no effort to optimise any of these parameters as we are not trying to optimise for accuracy, merely show our method performs well with a variety of architectures.

Each experiment here took less than two hours in total on a single machine with an NVIDIA GeForce RTX 2060 SUPER GPU and an AMD Ryzen 7 3700X 8-Core Processor. One run of the conformal prediction procedure has trivial overhead when compared with model fitting (and actually NAPS is faster than APS as we use less data points to calibrate the procedure).

