# OpenReview forum: "Distribution Free Prediction Sets for Node Classification"
_logconference.io/LOG/2022/Conference — LoG 2022 Poster_

### Official Review · Reviewer_SMab · 2022-10-12

**Overall Score:** 6
**Confidence:** 3

**Review:**

This paper proposes conformal prediction for graph-structure data, particularly node classification tasks. The authors first introduce the basic notion of exchangeable and non-exchangeable conformal prediction and propose 'NAPS' which is its application for node classification. Experiments on two datasets demonstrate that NAPS achieves the desired coverage and smaller size of average prediction sets than the previous method.

## Strong and weak points

The authors tackle a novel application of conformal prediction on node classification tasks. This setting is challenging since nodes in graphs violate the exchangeability assumption. The NAPS method leverages the homophily assumption of graphs, a common property of real-world graphs, and shows better results than the baseline using simple modification.

However, this paper has the following weaknesses.

- The paper is not easy to follow. This research is mainly about conformal prediction methods, but the target is graph-structured data. I am not claiming that this paper is not suitable for this venue. The GNN community (including myself) might not know much about conformal prediction, and an intuitive explanation for this seems necessary. It is not easy to see the big picture because the current manuscript is focused on technical details of conformal prediction.
- There is a discrepancy between the theory and the empirical results on the coverage gap. The NAPS shows the tighter bound in homophilous graphs (lines 89 – 91). However, although Flickr is a highly heterophilous dataset, NAPS seems to work practically well. It is necessary to explain whether the bound on the coverage gap is particularly meaningful or not.
- NAPS cannot be applied in transductive settings by nature (to use held-out data). The authors implicitly mentioned this by following the 'inductive node split' format. But I think the authors should clearly state that this method cannot be used in the transductive scenario.

## Recommendation

Based on the weaknesses above, I recommend a 'weak reject' for the current manuscript. However, I think these points can be revised for the revision period. I am willing to raise my score if the authors address my concerns.

## Questions and additional feedback

- What is $\delta$ in equation 2?
- Is there any relation between symmetric function and permutation invariant function? The former is used in this paper, and the latter is mainly used in the GNN community. It seems that these two are related; if it is, an explanation about these might help to understand the paper.
- What is the second form of randomness that needs to be controlled (lines 109 – 119)? The first is in the coverage, but I cannot find the second one.

---

### Official Review · Reviewer_zw8i · 2022-10-19

**Overall Score:** 8
**Confidence:** 2

**Review:**

## Summary
This paper proposes to estimate the predictive uncertainty of graph-structured data, especially for node classification tasks, based on prediction sets constructed from conformal prediction. In particular, unlike data in computer vision, every data instance (i.e., node) in graph-structured data is influenced by its relations to other data instances (i.e., neighborhoods). Thus, the authors utilize this property when calculating the prediction set by considering weights of the particular node's neighborhoods as 1. The experimental results show that the proposed model produces high-quality prediction sets compared to the baseline that does not utilize the graph property: structures of neighboring nodes.

---

## Strengths and Weaknesses

### Strengths
* The idea of considering the neighboring node structure for constructing prediction sets for graph-structured data is novel and interesting.
* The experimental results show that the proposed model produces more calibrated and tight prediction sets against the baseline.


### Minor Weaknesses
* When constructing the prediction set, the idea of simply setting the weight of k-hop neighborhoods for the particular node as all 1 is too naive, and there could be many different design choices (e.g., weighting based on the hop-size), which is not well developed.

---

## Minor Requested Changes
* It would be good to explore and include more analyses on the proposed weighting scheme of neighboring nodes.
* It would be interesting to see the analysis on varying the number of neighborhood hops (i.e., k).


---

## Questions
* In Line 100, why restricting the number of hop-sizes as 1 or 2? Is this because the computational efficiency issue when we consider more hops?

---

Note that I am not familiar with conformal prediction, thus I am willing to defend my rating that would be further influenced by other reviews.

---

### Official Review · Reviewer_a22H · 2022-10-20

**Overall Score:** 6
**Confidence:** 3

**Review:**

The paper is focused on extending conformal prediction for graph learning, with an example task of node classification. A method Neighborhood Adaptive Prediction Set (NAPS) is proposed which can be used to construct prediction sets for node classification, Section 3. NAPS leverages non-exchangeable conformal prediction with the given graph structure and can be used for node classification cases. NAPS is further empirically justified through experiments on two node classification tasks and is shown better compared to non-graph variant, Adaptive Prediction Set (APS). In the remainder of the paper (extended abstract-4 pages), the background of conformal prediction and its extension to result in NAPS is presented.

Pros:
1. The construction of prediction sets for node classification is encouraging, and straightforward extension based on recent works has been performed in the paper.
2. The integration of graph structure by taking into account the nodes' neighbors upto k hops in Eqn. 5 is intuitive.
3. Experiments show clearly that NAPS produces tighter prediction sets compared to APS that does not take graph structure into account.

Weakness and comments:
1. The paper mentions the case of inductive learning scenarios. However, it is demonstrated by single graph cases of Reddit2 and Flickr. How can the proposed method be applied to inductive scenarios of multiple (smaller) graphs where a trained model on one graph is transferred to another unseen graph for node prediction?
2. Follow up on point 1: the paper mentions as a limitation that lower quality prediction sets would be produced for nodes belonging to smaller connected components (eg. as observed in training scenarios of multiple smaller graphs). Perhaps a necessary discussion is missing that may elaborate this limitation of the method and its relevance to the broader/general 'inductive' node classification.
3. The experimental analyses is just limited to NAPS being better than APS which is positive. However, there is no further description or inference of the results in the paper; for eg. how are the NAPS scores observed across different models tested? are there any insights that could be useful to robustly understand the method?

Overall, the work and the method present in the paper is interesting; extends prior works in a convenient way; the contribution is limited which can be understood based on the paper's track. However, further questions may need to be addressed in how it is generally applicable on any node prediction tasks. Additionally, the experiments section could be better presented with additional insights.

---

### Official Review · Reviewer_s9vM · 2022-10-21

**Overall Score:** 6
**Confidence:** 4

**Review:**

The article "Distribution Free Prediction Sets for Node Classification" considers the problem of constructing prediction sets for node classification via conformal prediction methodology. The authors suggest to apply a recently proposed mechanism for construction conformal prediction sets in non-exchangeable case to the graph node classification. The resulting method was applied to two node classification problems with GNNs chosen as base classifiers. Experimental results show certain improvement of the proposed method over the baseline conformal prediction that fully ignores graph information.

Pros:
* The problem is relevant and application of conformal prediction to it is new to the best of my knowledge.

* Overall, the proposed approach is natural and the benefits of its application are clear.

Cons:
* The method is extremely basic and several extensions (mentioned by the authors) are almost on the shelf and worth trying.

* The experimental part is a bit simplistic with very few datasets being considered.

* The theoretical side of the paper is taken from the other work and essentially ignores the graph completely. It is of interest to develop the theoretical guarantee which would consider certain graph and feature generation mechanism.

To sum up, I think that the paper presents a simple yet interesting approach to improve coverage in conformal prediction for node classification. I think that many ways to extend and improve the paper exists, but being just an abstract of ongoing work it deserves to be presented at LoG conference as the topic is undoubtedly of interest for the community.

---

### Meta-Review · Area_Chair_LDse · 2022-11-16

**Confidence:** 4
**Recommendation:** Accept

**Meta Review:**

The authors extend conformal predictions to graph-structured data, where neighboring node structure is used for constructing prediction sets for inductive node classification tasks.

The work uses standard techniques adapted to a graph scenario. It is a good contribution overall. As one of the first works on this topic, it was good that the authors used the reviewers' feedback to improve the readability of the work (specifically for readers unfamiliar with conformal predictions). The authors did a good job with the rebuttal (e.g., including more experiments, addressing concerns)  and all reviewers were positive about the work and satisfied with the answers.

It is my opinion that this work should be accepted to LoG.

---

### Decision · Program_Chairs · 2022-11-22

Accept (Poster)